# Evolution of Caspases and the Invention of Pyroptosis

**DOI:** 10.3390/ijms25105270

**Published:** 2024-05-12

**Authors:** Betsaida Bibo-Verdugo, Guy Salvesen

**Affiliations:** 1Instituto Tecnológico de La Paz, Boulevard Forjadores de Baja California Sur 4720, La Paz 23080, Mexico; bet.bibo@gmail.com; 2Sanford Burnham Prebys Medical Discovery Institute, 10901 North Torrey Pines Road, La Jolla, CA 92037, USA

**Keywords:** caspase, evolution, pyroptosis, protein substrate, protein cleavage

## Abstract

The protein scaffold that includes the caspases is ancient and found in all domains of life. However, the stringent specificity that defines the caspase biologic function is relatively recent and found only in multicellular animals. During the radiation of the Chordata, members of the caspase family adopted roles in immunity, events coinciding with the development of substrates that define the modern innate immune response. This review focuses on the switch from the non-inflammatory cellular demise of apoptosis to the highly inflammatory innate response driven by distinct members of the caspase family, and the interplay between these two regulated cell death pathways.

## 1. The Caspase Protein Scaffold

Proteases are hydrolases that cleave peptide bonds between amino acids in proteins or peptides [1]. Proteolysis is a ubiquitous and irreversible posttranslational modification. When unspecific proteolysis occurs, proteins and peptides are degraded at multiple sites as can be found in the endo-lysosomal pathway and ubiquitin–proteasome system. Conversely, the limited and highly specific hydrolysis of proteins generates fewer cleavage products, often only two: and neo-N and -C termini. These newly generated fragments can yield proteins with new or modified functions or result in a loss of function. Thus, proteases govern signaling pathways and networks. Caspases constitute a family of cysteine proteases and have a stringent specificity for cleaving on the C-terminal side of Asp residues [2]. The caspase fold is sometimes also called the hemoglobinase fold because it is also found in legumains (hemoglobinases), paracaspases, metacaspases, and gingipains [3,4].

The hierarchical and structure-based protease classification by MEROPS defines the caspase-hemoglobinase fold as clan CD [5]. This clan encloses cysteine proteases with a distinctive α/β/α sandwich fold, for which the caspase-1 structure serves as the archetype (Figure 1). Members of the clan contain the catalytic dyad Cys and His for the nucleophilic attack of the scissile peptide bond [6]. Among the clan CD cysteine proteases, we find families of proteins with disparate evolutionary lineages, but several biochemical similarities. These are all endopeptidases (cleave peptide bonds of nonterminal amino acid residues in proteins), in contrast to exopeptidases that break bonds of terminal amino acids—and their specificity is dominated by a single amino acid in the substrate (the so-called P1 residue).

Clan CD encompasses the caspases with a specificity for cleavage directed to Asp, the paracaspases and metacaspases with a cleavage specificity directed to Arg, and legumains which cleave proteins after Asn residues were first identified in leguminous plants with homologues found later in animal parasites and fungi [8]. Other members of the CD clan include the B-cell activator MALT1 and clostripain—discovered in *Clostridum histoliticum* and both with a preference for hydrolysis after Arg [9,10]—and gingipain that is secreted by the bacteria *Porphyromonas gingivalis* with a specificity either for Arg or Lys and that contributes to the disease progression of gingivitis [11]. This broad repertoire of P1 specificities within clan CD underscores the plasticity inherent to the clan, likely due to the specificity-determining residues residing on surface loops in the zymogen (latent) forms [12]. One could speculate that this allows for variation without disrupting the core interactions in the catalytic domain. Thus, the unique fold of clan CD provides a malleable scaffold for the varying specificity to adopt new biological functions during evolution.

When considering the origin of the inflammatory caspases, it is instructive to recall that caspases required for apoptosis are found in all metazoan phyla investigated, from the earliest phylum (Cnidaria) through the nematodes and arthropods to vertebrates and mammals [13,14]. Based on their sequence identity and function, these proteases are either apoptotic or inflammatory. 

## 2. Unique Properties of Caspases

Structural and biochemical evidence shows that all caspases are obligate dimers in their active conformation [15]. The catalytic domain seen in caspases structures comprises a large (17–20 kDa) and a small (10–12 kDa) subunit, which are formed upon internal cleavages during caspase maturation at a segment that separates these subunits called the intersubunit linker. The large subunit contains the catalytic dyad residues. Proteolytic processing at the intersubunit linker stabilizes the catalytically competent conformation of caspases by creating contacts between the newly formed N- and C-termini of the large and small subunits from neighboring domains (Figure 1). Crystal structures show that, in their active conformation, the caspase catalytic domains interact, forming an almost perfect two-fold symmetry axis [7,16,17,18,19]. The small subunit of each monomer provides an interface that stabilizes the active conformation of the adjacent monomer [12]. Caspases play one of two roles in executing apoptosis: effector caspases must be activated by initiator caspases, thereby constituting the two-step process that helps to amplify the apoptotic stimulus and provide for control points [20].

A characteristic of the caspases involved in initiating apoptosis is the presence of N-terminal domains that direct them to oligomeric activation complexes in the cell, namely, the caspase activation and recruitment domains (CARDs) or death effector domains (DEDs) (Figure 2).

## 3. Activation Mechanisms

All caspases are obligate dimers in their active conformation; however, executioner (also known as apical caspases) and executioner caspases obtain their active state through different mechanisms (Figure 3). Apical caspases (caspase-8, -9, and -10) contribute with the first proteolytic signal leading to the initiation of apoptosis. Their latent forms (zymogens) are monomeric cytoplasmic proteins. These caspases contain N-terminal domains (CARDs or DEDs) that direct them to their activation modules, by means of homotypic connections to their interacting partner proteins Fas-associated via death domain (FADD) or apoptotic protease-activating factor 1 (APAF-1) [22,23]. The activation of apical caspases occurs via homo-dimerization forced by local concentration increments at their activation platforms, the death-inducing signaling complex (DISC), or the apoptosome [18,24,25,26]. Dimerization promotes a productive active site [18,25,27]. Cleavage alone is not sufficient for a full conversion into the catalytic form of these caspases [28].

Following the dimerization and generation of the first proteolytic signal, the initiator caspases are auto processed at the intersubunit linker with distinct mechanistic outcomes. The auto-cleavage of caspase-8 enhances the stabilization of its dimeric active conformation [29] and also aids in rearranging the active site architecture, inducing a slightly modified specificity for apoptotic substrates [30]. The activation of caspase-9 generates a neo-N-terminus that binds and leads to inhibition by X-linked inhibitor of apoptosis protein (XIAP). Caspase-3 further processes caspase-9 to remove a peptide containing the IAP-binding motif, thus releasing caspase-9 from XIAP inhibition and allowing the amplification of the apoptotic signal [31].

Apoptosis signaling is achieved via a two-step mechanism. First, initiator caspases are activated at their homo-dimerization platform. These active proteases then cleave and activate the zymogens of the apoptotic executioners (caspases-3 and -7) [23,32]. Executioner caspases are homodimers with a pre-formed active site in their latent conformation. In contrast to the apical caspases, a cleavage at the intersubunit linker of executioner caspases is enough to induce their activation and results in the re-arrangement of the substrate binding region loops, producing a catalytically viable active site [16,33]. The catalytic activity of caspase-3 is enhanced by more than four orders of magnitude as a result of proteolytic processing; thus, it constitutes the activation mechanism of executioner caspases [26].

The current model of the molecular activation of inflammatory caspases resembles that of the initiator caspases, where caspases are recruited via homotypic interactions between their CARD and interacting partners in multi-protein complexes known as inflammasomes [34,35]. Oligomerization at the inflammasomes induces a local increase in caspase-1 concentration, inducing its dimerization [36,37]. Like the apical caspases, self-processing at the intersubunit linker is required to generate fully active inflammatory caspases, to cleave protein targets and to exert their inflammatory effects [37,38,39]. Moreover, the processed intersubunit linker in inflammatory caspases constitutes a region of high affinity towards the binding of their main substrate—gasdermin D [40,41]. Therefore, like apical caspases, processing is required to produce cell death functional inflammatory caspases. Human caspase-4 and mouse caspase-11 have been proposed to act as direct intracellular sensors for lipopolysaccharide (LPS), which leads to their activation and, consequently, cell death [42,43,44]. However, the lack of known adaptor proteins in the activation of mouse caspase-11, or human caspase-4 and -5, leads us to question the molecular activation mechanism of these caspases.

## 4. Recognizing Substrates

As mentioned earlier, the defining property of the caspases is their stringent requirement for cleaving after Asp residues in substrates, governed by a deep and basic pocket (S1) shaped to accommodate the aspartic acid residue that precedes the cleavage site in the substrate (see Figure 4) [2,45]. The specificity for Asp at P1 is uncommon among proteases and only the serine protease granzyme B (a caspase activator) and the β1 subunit of the proteasome share this property with caspases [46]. The use of synthetic peptide libraries and proteomics demonstrates that pockets surrounding S1 fine-tune the specificity of individual caspases [2,45].

Methods to define substrate specificity that generate more conformational variability and longer peptide substrates than the canonical P1–P4 sites, such as phage display, revealed the importance of extended sequences in defining substrate specificity. In addition, exosites, regions of the enzyme that are distant from the catalytic site, participate in substrate binding and cleavage efficiency. The role of exosites in caspases is emerging as an important regulatory mechanism [41,49,50]. Hundreds of proteins have been identified as caspase substrates and curated in several databases, and the lists will undoubtedly grow as proteomics technology continues to advance [51,52,53]. Despite this wealth of data, only a handful of them have been verified to contribute to the morphologies of dying cells [2]. Moreover, caspase function goes beyond that of cell death signaling, as their proteolytic functions also modulate inflammation. Importantly, although many proteolytic events are required for apoptotic cell death, only a single cleavage of a signature protein is required to commit cells to an important inflammatory demise (see below).

## 5. Overview of Cell Death Executed by Caspases

Caspases are central modulators of the complex networks that take place during the execution of the three main death programs: apoptosis, necroptosis, and pyroptosis (outlined in Figure 5). The main morphological changes that accompany apoptosis include membrane blebbing, cell shrinkage, nuclear fragmentation, and, eventually, cell death [54]. In apoptotic cells, the cellular contents are constrained by apoptotic bodies, blebs of plasma membrane, preventing their scape. In vivo, apoptotic cells are rapidly recognized and cleared by macrophages’ ‘find me’ and ‘eat me’ signals, limiting the inflammatory effects that intracellular components may trigger when released to the extracellular environment [55]. In the past, necrosis was only recognized as a mere accidental form of cell death, i.e., induced by tissue trauma. However, our understanding of regulated necrosis is expanding, with newly described pathways of cell death characterized by cell swelling, cell membrane rupture, and the release of intracellular contents: pyroptosis and necroptosis [21]. When leaked into the extracellular environments, certain cellular contents have the potential to induce an inflammatory response by activating cells of the immune system. These molecules are called damage-associated molecular patterns (DAMPs). Hence, apoptosis is considered non-inflammatory, in contrast to pyroptosis and necroptosis which promote an inflammatory environment.

## 6. Overview of Apoptosis

Apoptosis has been observed throughout the animal kingdom, but its execution in invertebrates differ to the pathways described here [56]. Apoptosis is initiated by two main pathways in vertebrates, extrinsic and intrinsic apoptosis. Intrinsic (or mitochondrial) apoptosis is initiated with the activation of caspase-9, while the activation of caspase-8 activates the death receptor (or extrinsic) pathway (Figure 5). For a detailed description of these pathways, consult [57,58].

The assembly of the apoptosome during intrinsic apoptosis can be triggered by multiple stressors. These stressors converge on the permeabilization of the mitochondrial outer membrane, which results in the diffusion of mitochondrial proteins into the cytosol. Mitochondrial protein cytochrome-C then signals the formation of the apoptosome, which also contains the cytoplasmic protein Apaf-1. The recruitment and activation of caspase-9 occur in this complex [59].

The DISC is formed upon the ligation of death receptors (DRs). These proteins are members of the tumor necrosis factor (TNF) superfamily, that includes the tumor necrosis factor receptor-1 (TNFR1). These receptors contain a cytoplasmic region known as the death domain that allows the receptor to engage in cell death signaling once activated by its cognate ligand. Upon ligand binding, the DR receptor, in co-operation with adaptor proteins, forms the DISC. The DISC is a molecular platform where caspase-8 is recruited and activated [60,61].

TNFR1 provides a critical control point at the nexus of cell death and survival (Figure 5). Depending on the relative expression of intracellular signaling proteins, the ligation of this receptor may engage inflammatory signaling by the NF-κB route, apoptosis by the extrinsic pathway, or necroptosis through the engagement of receptor interacting protein kinase 1 (RIPK1). It may seem bizarre that a single apical caspase (caspase-8) can regulate the switch of cell death mechanisms from a non-inflammatory outcome (apoptosis) to an inflammatory one (necroptosis). But we find it even more curious that the caspase locus has evolved away from an apoptotic role to a clearly major inflammatory role through the development of the inflammatory caspase locus.

The executioner caspases are the main substrates of the initiator caspases, thus providing a proteolytic activation cascade that culminates in the execution of apoptosis [21]. Hundreds of protein substrates have been linked to executioner caspases, but it is complex to point out which of these are required for cell death completion. In the view of the non-inflammatory consequences of apoptosis, one could argue that caspase-cleavage may serve dying cells by depleting potentially inflammatory intracellular components [55]. 

Upon their activation, executioner caspases cleave hundreds of proteins simultaneously; these proteolytic events culminate in the strategic demise of the cell where the cellular contents are restricted by the membrane. In contrast, a single proteolytic event is required for the induction of pyroptosis [62,63]. The cleaved gasdermin D is responsible for the distinctive lytic phenotype of pyroptosis. In addition, inflammatory caspases have developed the ability to convert members of the interleukin-1 family members, mainly prointerleukin-18 and -1β (pro-IL18 and pro-IL1β), to obtain the ability to interact with their cognate receptors. The interleukin precursors lack a signal peptide, and their secretion occurs via gasdermin D membrane pores during pyroptosis.

## 7. Split of the Inflammatory Caspase Branch from the Main Caspase Trunk

Pyroptosis is a caspase-dependent paradigm of cell death with important implications for the innate immune response and clearance of pathogens [64]. The process is defined by the disruption of the cell membrane by the cleaved gasdermin D. Gasdermin D cleavage is mediated by inflammatory caspases; caspase-1, -4, and -5 in human, and caspase-1 and -11 in mouse. Inflammatory caspases are activated in protein complexes known as inflammasomes. The term was originally coined by Martinon et al. in a pivotal article describing the assembly of these structures [65]. An inspection of the evolutionary history of gasdermin D suggests that pyroptosis first appeared in Mammalia [56,66,67]. However, other pore-forming members of the gasdermin family are found in invertebrates [66]. An interesting example is found in corals, where a gasdermin E homologue may be involved in mediating pathogen-induced coral death [68].

Gasdermin D is found in a latent form in cells due to auto-inhibition interactions between its N-terminal lytic domain and its auto-inhibitory C-terminal domain. The proteolytic processing of gasdermin D by inflammatory caspases at a conserved position (Asp276 in mouse and Asp275 in human) liberates the N-terminal domain [62,63,69]. The N-lytic fragment translocates and oligomerizes in the plasma membrane, forming pores and, ultimately, causing the characteristic lytic phenotype of pyroptosis [70,71,72]. Gasdermin D forms pores of an approximate diameter of 10–30 nm in liposomes [69,70,71,72,73]. Membrane rupture also depends on a protein called Ninjurin-1, by a mechanism yet to be clarified [74,75], ultimately resulting in the release of most cytosolic proteins, but with the retention of organelles within the cell corpse [76]. In the absence of recognized secretion signals, a gasdermin-dependent membrane rupture provides for the release of biologically active IL18 and IL1β [77,78,79,80].

Two pathways leading towards the pyroptotic outcome are recognized (Figure 5), distinguished by the extracellular vs. intracellular recognition of pathogens, and by which caspases execute the pathway. The canonical pathway is triggered by the recognition of extracellular pathogens, primarily by Toll-like receptors (TLRs) leading to the activation of caspase-1 via clustering with intracellular pattern recognition receptors (PRPs) such as nucleotide-binding oligomerization domain (NOD), leucine-rich repeat (LRR)-containing protein (NLR) family members (NLRP1, NLRP3, and NLRC4), and the proteins absent in melanoma 2 (AIM2) and pyrin [34]. The oligomerization of these proteins and recruitment of adaptor protein ASC leads to the formation of the canonical inflammasome, where caspase-1 is activated in a mechanism somewhat reminiscent of the apoptosome of intrinsic apoptosis [65,81]. While the canonical inflammasome involves a variety of stimuli and proteins, the only known inducer of the non-canonical pathway is when lipopolysaccharide (LPS) from Gram-negative bacteria reaches the cytosol [42,43]. Caspase-1 is not involved in this process; rather, Caspase-4 and -11 have been proposed to act as direct intracellular sensors for lipopolysaccharide, leading to their activation and, consequently, pyroptosis, thus defining the non-canonical pathway [42,43,44].

Pro-IL18 is constitutively expressed by most cells whereas pro-IL1β is expressed primarily by hematopoietic cells [82]. These important inflammatory cytokines are the only interleukin-1 (IL1) family members that require proteolytic processing to gain their biologically active conformation [83,84,85,86]. In vivo studies in mice, backed up by biochemical studies, suggest that IL18 and IL1β conversion into the bioactive cytokine is performed primarily by caspase-1 to produce the bioactive cytokine [42,48,87]. The activation of these cytokines by caspase-11 is very inefficient [48,87]. Rather, the NLRP3 inflammasome is activated downstream of non-canonical pyroptosis, resulting in the activation of the cytokines in a caspase-1-dependent manner [42,88]. Hence, in both the canonical and non-canonical inflammasomes, the cleavage of IL18/IL1β and gasdermin D occur simultaneously, providing us with a mechanism for cytokine activation and release. Importantly, organelles are retained in the cell corpse during the course of pyroptosis [76]. An additional layer of complexity in the regulation of cytokine activation by inflammatory caspases surges from the cleavage at alternative sites [89].

## 8. Co-Evolution of Caspases and the Pyroptotic Substrates

Invertebrates, ranging from protozoans to metazoans, have an arsenal of tools that allows them to distinguish the self from the non-self. This set of mechanisms includes, but is not limited to, pattern recognition receptors that identify well-conserved pathogen molecular patterns, effector molecules and cells, and signaling molecules, like cytokines [90]. The innate immune system aims to secure survival and to maintain homeostasis against many different types of pathogens. Primitive invertebrates presented the first signs of adaptive immunity [91]. When adaptive immunity evolved in the vertebrate lineage, many of the pre-existing elements of innate immunity were incorporated in the new host to improve and optimize immunity against the new challenges. In a nutshell, the sensing mechanisms of the innate immune system guide the activation of suitable effector responses of the adaptive immune system [92].

Pyroptosis, recognized as the inflammatory cell death paradigm elicited by a gasdermin D lytic fragment, is an important mechanism that links the innate and adaptive immune systems, evolved in the vertebrate lineage. The origin of pyroptosis, and which emerged first, the cleavage of Gasdermin D or pro-inflammatory cytokines, is elusive, given how little is known with clarity about the co-evolution of caspases and their substrates. Parsimony would suggest that the cleavage of Gasdermin D must have emerged first because, even though pro-inflammatory cytokines may be cleaved, there is no strategy for them to exit the cell unless it has first undergone a lytic event. However, an inspection of protein sequences suggests that caspase-1, IL1β, and IL18 preceded gasdermin D and appeared simultaneously with adaptive immunity early in the evolution of Vertebrata (Figure 6).

Gasdermin family members can be found in early metazoans like Cnidaria and Mollusca [66,68]. The family probably expanded in a common ancestor of the jawed vertebrates, perhaps allowing for a more diverse set of mechanisms of activation and cell membrane permeabilization functions among vertebrates [66,69]. Gasdermin D was present in a common ancestor of Mammalia and the cleavage site that generates the lytic fragment is prevalent in placental mammals (Figure 6). Similarly, the consensus cleavage site sequence of IL1β that generates the mature and active cytokine first appeared in mammals (Figure 6). This suggests that the cargo, the biologically active IL1β, appeared simultaneously with the gate, the gasdermin D pores. A big question that arises is the gap between caspase-1 evolution and gasdermin D, which suggests earlier caspase-1 functions that could be disentangled by investigating its role among non-mammalian vertebrates. In addition, there might be caspase-1/gasdermin D-independent routes of IL1β activation and release among these vertebrates.

A persistent question in the field is the diversity of inflammatory caspases in different groups of mammals. While most mammals express a version of caspase-4, caspase-5 is only found in higher primates (Figure 6). These enzymes are considered synonymous homologues of mouse caspase-11, and this would imply that caspase-4 and -5 would be functionally interchangeable. However, the expression profiles and functional biochemistry experiments indicate that this might not be the case [48,93]. Evidence suggests that caspase-5 might be a closer functional homologue to caspase-11 than caspase-4. First, caspase-5 is under the regulation of an NF-κB promoter [94]. Therefore, the *CASP5* expression is highly upregulated by LPS [95,96]. Of course, further exploration is needed to establish a functional equivalence.

In addition to inflammatory caspases, apoptotic caspase-3 is also able to cleave gasdermin D. However, this event fails to generate the functional cell lytic fragment. Instead, caspase-3 cleaves gasdermin D at an alternative site (Asp87) within this fragment, frustrating the pyroptotic function of this protein [97]. Hence, when apoptosis ensues, caspase-3 provides us with a mechanism to prevent unwanted inflammation through the inactivation of gasdermin D. This suggests a strong co-evolution of the caspase landscape, both apoptotic and inflammatory, with the pyroptotic substrates. Factors like the relative availability of active caspase-1 and caspase-3 during pyroptosis and apoptosis may be defining to the cell death fate.

**Figure 6 ijms-25-05270-f006:**
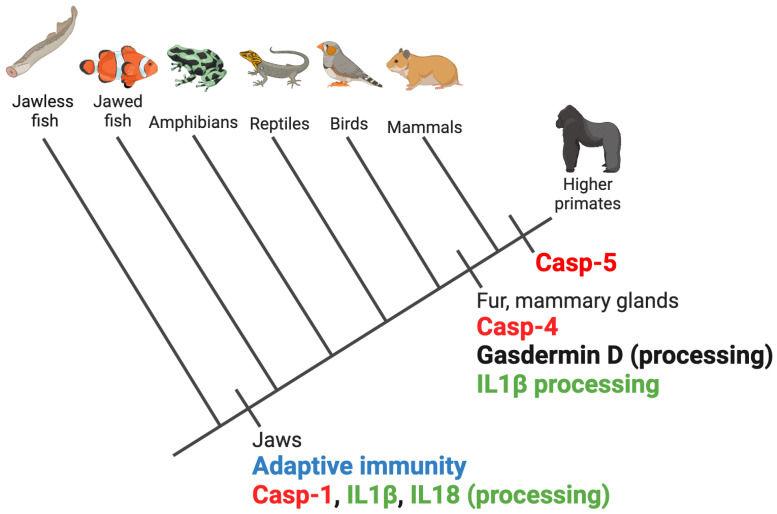
Representation of the phylogeny of vertebrates and the predicted appearance of inflammatory caspases, IL1β, IL18, and gasdermin D. Protein sequences were searched by using the tblastn program of the BLAST server [98]. The occurrence of proteins was predicted by protein conservation across different groups of mammals and cleavage site was searched by alignment and inspection of protein sequences of representative of each group of vertebrates.

## 9. Pseudocaspases (cFLIP and Caspase-12)

### 9.1. cFLIP

Adjacent to the caspase-8 locus is an inactive homologue known as *cFLIP*. This gene is expressed in two alternative spliced forms: a short version (known as *cFLIP_S_*) and a large version (*cFLIP_L_*) [99]. *cFLIP_S_* codes a truncated protein that only contains the recruitment domains and can block the activation of caspase-8, forming an inactive heterodimer. In contrast, cFLIP_L_ is a pseudocaspase with the overall fold of caspase-8 but is devoid of activity due to mutations of catalytic residues. Thus, cFLIP_L_ forms an active heterodimer with caspase-8. This heterodimer has a different substrate specificity and function than the homodimeric caspase-8, which has an apoptotic role [100] (Figure 5).

The heterodimer formed between caspase-8 and cFLIP_L_ is catalytically productive and acts as a molecular switch between the engagement of apoptosis and inhibition of necroptosis [101]. Three concepts are critical in understanding this switch: the relative concentrations of caspase-8 and cFLIP_L_ [102], the cell type specificity [103], and the substrates that are targeted differentially by the caspase-8 homodimer vs. the caspase-8/cFLIP_L_ heterodimer [100]. Which substrates are cleaved preferentially by the caspase-8/cFLIP_L_ heterodimer to prevent necroptosis, and the molecular mechanism that alters caspase-8 specificity when forming a heterodimer with cFLIP_L_ are questions that need assessment.

Several known caspase-8 substrates have roles in necroptosis: caspase-8 itself, cFLIP_L_, RIPK1, RIPK3, and the RIPK1-deubiquitylating enzyme cylindromatosis (CYLD) [100,104,105,106]. The current model suggests that the cleavage of key proteins of the necroptosis pathway by the caspase-8/cFLIP_L_ heterodimer results in their inactivation, thereby negatively regulating necroptosis. Caspase-8-deficient mice are nonviable due to the survival function of the caspase-8/cFLIP_L_ heterodimer [101]. Mice expressing an uncleavable version of caspase-8, cFLIP_L_, or CYLD show no difference in viability compared to WT animals, showing that these are not the essential caspase-8 substrates during embryogenesis [107]. A critical caspase-8 substrate is RIPK1; the cleavage at Asp325 in RIPK1 is crucial for preventing aberrant cell death during development and TNF-induced cell death in vitro [107,108]. A likely mechanism is that the cleavage of RIPK1 by caspase-8 results in the disassembly of cell death signaling complexes [107].

### 9.2. Caspase-12

A homology analysis classifies caspase-12 as an inflammatory caspase [36]. However, the function of this protein in mammals is still unclear. In the majority of the human population, *CASP12* is a pseudogene encoding a premature stop codon that results in rapid mRNA degradation, which impedes the transcription of this protein [109]. A full-length allele can be found in a fraction of people of African descent with a 20% frequency [110]. Nevertheless, a functional protein has not been identified in this population and an additional mutation in the SHG box, a critical site for catalytic function in caspases, allows us to predict that this protein lacks catalytic activity [111]. Conversely, rodents encode a full-length *CASP12* gene. When recombinantly expressed, rat caspase-12 can auto-process, suggesting that this protein may be catalytically competent, although this remains to be confirmed in vivo [112]. In mice, caspase-12 is expressed as a full-length protein but presents a substitution in a residue that forms part of the main specificity pocket in caspases and defines their preference for P1-Asp (Arg341 to Lys). Thus, this evidence predicts that caspase-12 is not a functional caspase. One of the proposed functions for caspase-12 is as a negative modulator of caspase-1 activity, analogous to that of the caspase-8/cFLIP_L_ in the regulation of pyroptosis [113]. However, such conclusions must be reinterpreted under the light of new experiments because they were originally performed on a knockout strain that was also deficient in caspase-11 [42,114]. Indeed, the same caspase-11 inactivating mutation exists in many mouse models, clouding the interpretation of the role of proposed inflammatory mediators based on the 129 mouse genetic background [115]. For example, this mutation misguided the interpretation of the caspase-11 role in septic shock models [42]. In conclusion, biochemical data suggest that caspase-12 is a pseudocaspase that may act as a modulator of inflammatory caspases, but this needs further inspection.

## 10. Variation of Inflammatory Caspases: Evading Pathogens

Typically, *CASP1* and *CASP4* are contiguous genes, except for in primates where *CASP5* is between these two genes [116]. Interestingly, the genome of modern mammals of the order Carnivora contains a deletion in the CARD of caspase-1, resulting in a read-through to create a caspase-1/caspase-4 hybrid gene. This gene codes for two alternative spliced variants which vary in their CARDs (caspase-1/-4), but both contain the catalytic domain of caspase-4 [116,117]. Thus, Carnivora lack the caspase-1 catalytic domain but express an unusual version of caspase-4 that can activate both gasdermin D and IL-1β [117,118]. An ancestral sequence reconstruction of a putative ancestor of the Carnivora inflammatory caspase revealed that it activates gasdermin D but has a reduced ability to activate IL-1β, suggesting that caspase-1 was lost in a Carnivora ancestor, perhaps upon a selective pressure for which the generation of biologically active IL-1β by caspase-1 was detrimental [118]. Later, a member of Carnivora encountered selective pressures that required the production of IL-1β, and caspase-4 subsequently gained this activity. This hypothesis would explain why existing Carnivora possess an inflammatory caspase with caspase-1-like specificity but placed on a caspase-4 scaffold [118].

The lack of a caspase-1 orthologue in Carnivora suggests that deletion occurred early in the evolution of these mammals. The black queen hypothesis, in an analogy to the card game Hearts, postulates that certain genes or biological functions are costly or undesirable [119]. Hence, gene deletion or loss-of-function comes to an advantage and may represent an evolutionary adaptive response if the loss outweighs the cost [119,120]. In this context, we reason that the absence of caspase-1 suppressed detrimental immune responses perhaps associated with a microbe-rich diet or noxious infections suffered by a Carnivora ancestor. An alternative explanation may be that caspase-1 became dispensable. However, the later scenario is less likely since several other genes involved in inflammatory cell death are absent or are present as pseudogenes in this clade [117,121].

## 11. Concluding Remarks

Inflammatory caspases are central to the mechanisms governing innate immune responses. By integrating the surveillance tools of PRPs with the activity of caspases, a quick response can be mounted against numerous threats via pyroptotic cell death. Here, we exposed that inflammatory caspases are more recent additions to the cell death machinery than apoptotic caspases. Inflammatory caspases co-evolved with their main substrates during the history of Chordata, leading to the pyroptotic pathway. Nevertheless, many questions deserve further exploration, including the mechanism of ninjurin-1 in mediating plasma membrane rupture during cell death. Moreover, while caspase-1, IL18, and IL1β appeared approximately simultaneously early in the evolution of vertebrates, gasdermin D and IL1β processing are additions observed later in mammals. This begs the question of how IL1β is activated and released in other branches of vertebrates, like fish and birds. A case of controversy is the differential function of caspases-4 and -5 in primates. Of particular interest is if they specialize in responses against intracellular pathogens like bacteria or viruses, and if they are assisted by other proteins in those responses. Clarifying this question is important to establish the translation of mouse studies to human disease. Caspase-12 had some momentum but later declined in attention. Based on the biochemical data, we recommend that caspase-12 should be considered a pseudocaspase. Of particular interest is the function of caspase-12 in the fraction of people of African descent that carry the read-through gene.

## Figures and Tables

**Figure 1 ijms-25-05270-f001:**
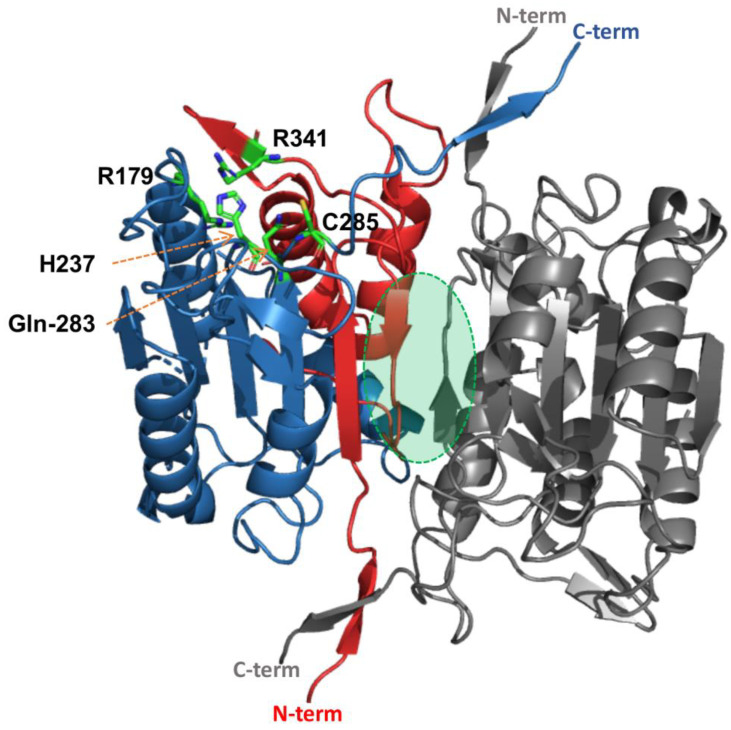
The crystal structure of human caspase-1 exemplifies the fundamental caspase fold. PBD entry: ICE1 [7]. Caspases are homodimers formed by monomers with large (blue) and small (red) subunits, which are originated from a single-chain protein by proteolysis. The caspase-fold consists of a 12-stranded β-sheet that is sandwiched by α-helices. The catalytic residues (His-237 and Cys-285) and residues forming the specificity pocket (Arg-179, Gln-283, and Arg-341) that accommodate the Asp-P1 of the substrate during catalysis are represented by green sticks. Most of the dimer contact area is built by the central small subunits (indicated by green shade), with additional interactions between the C- and N-termini of large and small subunits.

**Figure 2 ijms-25-05270-f002:**
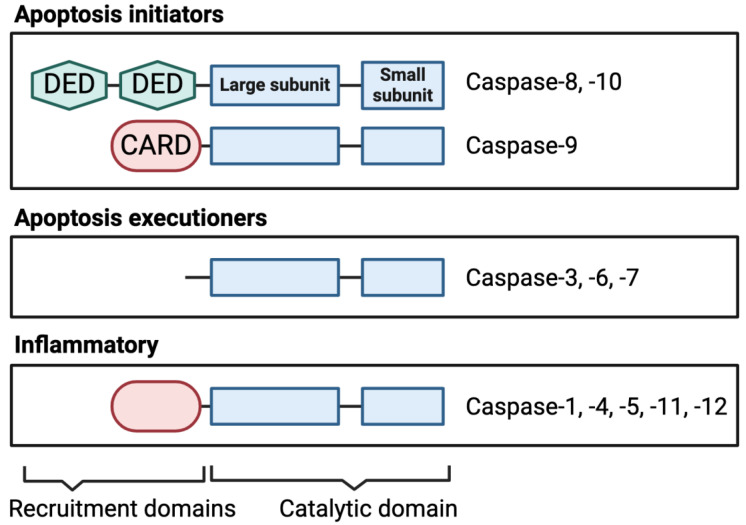
Caspase structural domains. Apoptotic caspases are classified according to their role in cell death signaling in initiators and executioners. The initiator caspases contain N-terminal death effector domains (DEDs) or caspase activation and recruitment domains (CARDs), which direct them to their activation platforms via interaction with adaptor proteins. Inflammatory caspases are those involved in pyroptosis signaling and contain a CARD similarly to the apoptosis initiator caspase-9. In their mature, active forms, the caspase catalytic domain (in blue) contains one large and one small subunit. Caspase-2 and -14 are also present in humans but they are not known to be involved in cell death or the evidence is contradictory [21].

**Figure 3 ijms-25-05270-f003:**
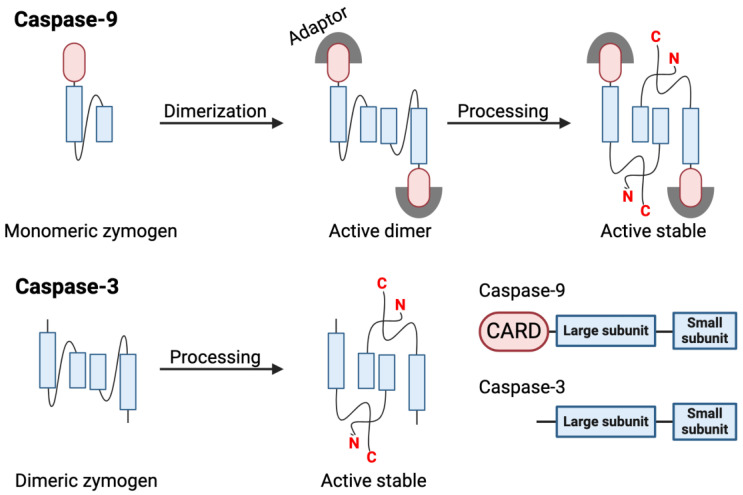
Caspase activation mechanisms. Caspase activation mechanisms are best understood for apoptotic caspases, represented here for illustration. Apical caspase-9 contains a CARD (pink) that directs it to activation platform (the apoptosome) via interaction with adaptor proteins (gray). The catalytic domain (blue) contains the large and small subunits. In their latent conformation, apical caspases are monomeric proteins. These caspases are activated via induced-dimerization induced by increments in concentration at their activation platforms. The fully active death-inducing conformation is only attained by auto-proteolytic processing at the intersubunit linker. Caspase-3 is the model of the activation mechanism of the executioner caspases, which are already dimeric in their latent state, and require proteolytic processing at the intersubunit linker to gain their active conformation.

**Figure 4 ijms-25-05270-f004:**
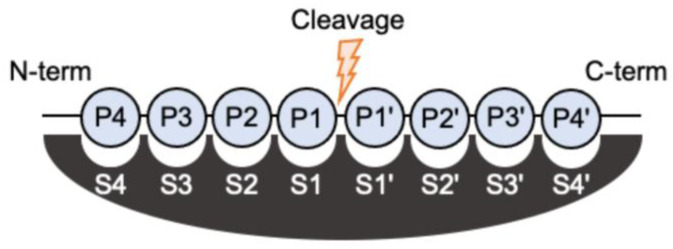
Nomenclature of substrate residues interacting with a protease active site. The amino acid of the N-terminal side relative to the cleaved peptide bond is named the P1 residue, while that on the C-terminal side is the P1’ residue. The consecutive residues in the primary structure of the substrate are numbered consecutively, taking the cleaved peptide bond as the origin [47]. Substrate residues form specificity pockets, defined by the protease sequence and structural features, that interact with the substrate residues. These sites are named with an S and are numbered following the same logic used for the substrate. Adapted from [48].

**Figure 5 ijms-25-05270-f005:**
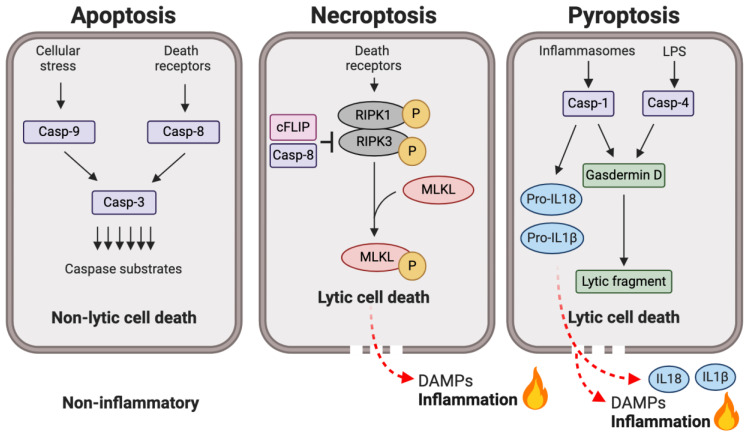
The role of caspases in the regulation of cell death pathways. This diagram represents the distinguishing caspase mechanisms in three paradigms of cell death. Apoptosis and pyroptosis are signaled by different sets of caspases; while apoptotic caspases are involved in apoptosis, pyroptosis engages inflammatory caspases. The main substrate of apical caspases-8 and -9 are the executioner caspases, i.e., caspase-3, which cleave multiple proteins, resulting in cellular decay. The key event triggering necroptosis is phosphorylation of MLKL by RIP kinases (RIPK1 and RIPK3). This cell death modality is negatively regulated by a heterodimer formed by caspase-8 and cFLIP. Cleavage of gasdermin D by inflammatory caspases is critical for engaging pyroptosis. Necroptosis and pyroptosis are considered inflammatory because they allow the release of damage-associated molecular patterns (DAMPs), which exert their mechanisms on the innate immune system. During pyroptosis, inflammation may be amplified by conversion and release of inflammatory cytokines IL18 and IL1β.

## Data Availability

All protein alignments are available from the authors upon request.

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
