# Peer review of "Evolution of Caspases and the Invention of Pyroptosis"

_ijms, 2024, doi:10.3390/ijms25105270_

Round 1

Reviewer 1 Report

Comments and Suggestions for Authors

Dear editorial team of IJMS,

the present review of Bibo-Verdugo and Salvesen about the role of caspases in regard to cell-death and especially pyroptosis is well organized and very readable. All aspects given in the abstract are addressed in the main text. Although a number of reviews exist which summarize features of caspase properties, the present work is very helpful for a broad readership. Expecially the evolutionary context is interesting and rarely discussed in other reviews.  I was pleased by the figures which are very supportive. This holds  true for flagging caspase function even in invertebrates.

Taken together, I would recommend this review for publication in your esteemed journal.

Best regards

Author Response

the present review of Bibo-Verdugo and Salvesen about the role of caspases in regard to cell-death and especially pyroptosis is well organized and very readable. All aspects given in the abstract are addressed in the main text. Although a number of reviews exist which summarize features of caspase properties, the present work is very helpful for a broad readership. Expecially the evolutionary context is interesting and rarely discussed in other reviews.  I was pleased by the figures which are very supportive. This holds  true for flagging caspase function even in invertebrates.

Taken together, I would recommend this review for publication in your esteemed journal.

We appreciate the kind comments of Reviewer 1.

Reviewer 2 Report

Comments and Suggestions for Authors

The review article addresses a topic of interest to the scientific community, mainly the participation of caspase-1 and 4 in pyroptosis. The article is well structured and clearly presented, reaching the proposed theme "Evolution of Caspases and the invention of Pyroptosis"

The lack of line numbering makes it difficult to point my suggestions

1- Some statements require references, despite being classic information. For example, the first paragraph of the introduction:

Proteases are hydrolases that cleave peptide bonds between amino acids in proteins or peptides. Proteolysis is a ubiquitous and irreversible posttranslational modification. When unspecific proteolysis occurs, proteins and peptides are degraded at multiple sites as can be found in the endo-lysosomal pathway and ubiquitin-proteasome system. Conversely, limited and highly specific hydrolysis of proteins generates fewer cleavage prod-ucts, often only two: and neo-N and -C termini. These newly generated fragments can yield proteins with new or modified functions or result in loss-of-function. Thus, proteases govern signaling pathways and networks.

Suggestion: Alan J. BARRETT and J. Ken McDONALD, Nomenclature: protease, proteinase and peptidase, 1986

2- Please correct the Legend of Figure 2

3- Top of page 5

Auto cleavage of caspase-8 enhances the stabilization of its dimeric active conformation [28] And also aids in rearranging the active site architecture inducing a slightly modified specificity for apoptotic substrates [29].

4- Recognizing Substrates

The specificity for Asp at P1 is very rare among proteases and only the serine protease granzyme B (a caspase activator) and the β1 subunit of the proteasome, share this property with caspases [45].

This statement is not correct. The Merops platform indicates that several proteases (of various classes) can cleave substrates with P1= Asp.

Suggestion: Replace "very rare" by uncommon.

5- Overview of cell death executed by caspases

Caspases are central components of the complex biochemistry that takes place during the regulation and execution of the three main death pathways: apoptosis, necroptosis and pyroptosis (these pathways are outlined in Figure 5).

Suggestion: Caspases are central components of the complex biochemistry pathways that takes place during the regulation and execution of the three main death pathways: apoptosis, necroptosis and pyroptosis (these pathways are outlined in Figure 5).

Comments on the Quality of English Language

Minor editing of English language required.

Author Response

The review article addresses a topic of interest to the scientific community, mainly the participation of caspase-1 and 4 in pyroptosis. The article is well structured and clearly presented, reaching the proposed theme "Evolution of Caspases and the invention of Pyroptosis".

The lack of line numbering makes it difficult to point my suggestions.

1- Some statements require references, despite being classic information. For example, the first paragraph of the introduction:

Proteases are hydrolases that cleave peptide bonds between amino acids in proteins or peptides. Proteolysis is a ubiquitous and irreversible posttranslational modification. When unspecific proteolysis occurs, proteins and peptides are degraded at multiple sites as can be found in the endo-lysosomal pathway and ubiquitin-proteasome system. Conversely, limited and highly specific hydrolysis of proteins generates fewer cleavage products, often only two: and neo-N and -C termini. These newly generated fragments can yield proteins with new or modified functions or result in loss-of-function. Thus, proteases govern signaling pathways and networks.

Suggestion: Alan J. BARRETT and J. Ken McDONALD, Nomenclature: protease, proteinase and peptidase, 1986

We added the reference as suggested.

2- Please correct the Legend of Figure 2

We don’t understand this comment. We are not able to notice anything wrong with the figure legend.

3- Top of page 5

Auto cleavage of caspase-8 enhances the stabilization of its dimeric active conformation [28] And also aids in rearranging the active site architecture inducing a slightly modified specificity for apoptotic substrates [29].

This comment eludes us. We are not able to notice anything wrong with the text.

4- Recognizing Substrates

The specificity for Asp at P1 is very rare among proteases and only the serine protease granzyme B (a caspase activator) and the β1 subunit of the proteasome, share this property with caspases [45].

This statement is not correct. The Merops platform indicates that several proteases (of various classes) can cleave substrates with P1= Asp.

Suggestion: Replace "very rare" by uncommon.

We hear what the reviewer says, and we followed their recommendation.

5- Overview of cell death executed by caspases 

Caspases are central components of the complex biochemistry that takes place during the regulation and execution of the three main death pathways: apoptosis, necroptosis and pyroptosis (these pathways are outlined in Figure 5).

Suggestion: Caspases are central components of the complex biochemistry pathways that takes place during the regulation and execution of the three main death pathways: apoptosis, necroptosis and pyroptosis (these pathways are outlined in Figure 5).

We have modified the text as follows: “Caspases are central components of the complex networks.”

Comments on the Quality of English Language

Minor editing of English language required.

We revised the manuscript and modified accordingly.

Reviewer 3 Report

Comments and Suggestions for Authors

The topic of the review is interesting and well presented but some improvements are required as follows:

1. The abstract is too short, add some information.

2. I suggest to add a paragraph in "Recognizing substrate" about some specific substrates for caspases.

3. You have to illustrate the detailed involvement of caspase in apoptosis by a graphical presentation to make it easier for the readers.

4. I strongly suggest to add the breakthrough discovery of pyroptosis in "Split of the inflammatory caspase branch from the main caspase trunk".

5. In "Casp-12" heading, you added information about it from rats and mice. What about in humans? There are a good literature about that. Please update.

6. Uniform your writing way for caspases i.e. write all in abbreviation or write all in full name.

7. "Coevolution of caspases and the pyroptotic substrates" should be directly after "Split of the inflammatory caspase branch from the main caspase trunk".

8. In Fig. 6, Protein sequences were searched by using the tblastn program of the BLAST server [112]. you have to add such details in the main text to describe what are these proteins, from where, for what purpose?

9. Add a conclusion at the end to strength your review.

10. Try to add some references from 2024 to be up-to-date.

Author Response

The topic of the review is interesting and well-presented, but some improvements are required as follows:

  1. The abstract is too short, add some information.

We disagree. In our opinion an abstract of a review article should give enough information to draw readers in, providing the central thrust of the review. no more than this.

  1. I suggest to add a paragraph in "Recognizing substrate" about some specific substrates for caspes.

We understand this request, however the field is replete with data derived from proteomic experiments and we hesitate to pronounce on which proposed cleavages are important for a full apoptotic outcome, hence we have re-phrased this section very carefully.

  1. You have to illustrate the detailed involvement of caspase in apoptosis by a graphical presentation to make it easier for the readers.

Figure 5 includes the salient framework of caspases in apoptosis pertinent to the scope of this review. We sought to highlight the role of caspases in inflammation, not to provide a detailed view of the protein network of apoptosis. To guide readers to more comprehensive and detailed overviews of this topic we have referenced comprehensive reviews.

  1. I strongly suggest to add the breakthrough discovery of pyroptosis in "Split of the inflammatory caspase branch from the main caspase trunk".

We added an extra sentence and reference.

  1. In "Casp-12" heading, you added information about it from rats and mice. What about in humans? There are a good literature about that. Please update.

We disagree with this point. We began this section with a description of the human protein with information that we consider relevant about caspase-12.

  1. Uniform your writing way for caspases i.e. write all in abbreviation or write all in full name.

To address this suggestion, we Changed casp to caspase throughout the text.

  1. "Coevolution of caspases and the pyroptotic substrates" should be directly after "Split of the inflammatory caspase branch from the main caspase trunk".

We are made the suggested change.

  1. In Fig. 6, Protein sequences were searched by using the tblastn program of the BLAST server [112]. you have to add such details in the main text to describe what are these proteins, from where, for what purpose?

This is a review article, and we are trying to keep it available and enjoyable for a wide audience. Anybody who wants to see the full alignments can email the authors.

  1. Add a conclusion at the end to strength your review.

Moving the sections around allowed us to include a concluding paragraph, which we have added to the paper.

  1. Try to add some references from 2024 to be up-to-date.

To address this suggestion we have updated the reference section.